# Lightweight Design of Shock-Absorbing and Load-Bearing Components Based on 3D Printing Technology

Guoqing Zhang [1,]*[ID], Rongrui Feng [1], Junxin Li [1], Yongsheng Zhou [1], Xiaoyu Zhou [1] and Anmin Wang [2]

1   School of Mechanical and Electrical Engineering, Zhoukou Normal University, Zhoukou 466000, China; hy13903875385@163.com (R.F.); lijunxin1995@163.com (J.L.); zhouyongsheng1@sina.com (Y.Z.); vzhouxy@163.com (X.Z.)
2   School of Mechanical and Automotive Engineering, South China University of Technology, Guangzhou 510640, China; wanganminhnlg@163.com
*   Correspondence: zhangguoqing1202@sohu.com

**Abstract:** Nowadays, the redesign of new shock-absorbing load-bearing parts has gradually gained more and more focus due to the pressure of energy, environmental protection, and people's pursuit of high-performance (light weight, excellent shock absorption, etc.) travel tools, and the development of 3D printing technology provides the possibility to design such high-performance parts. Therefore, firstly, the strength analysis of the parts is carried out by adopting Altar Inspire software, then topology optimization design is conducted in Inspire software and, finally, direct manufacturing is carried out using Aurora 3D printers. The results show that the maximum Mises equivalent stress of the shock-absorbing load-bearing components after lightweight design is not more than the material's yield stress of 45 MPa and the safety factor (1.5) is greater than the minimum allowable safety factor (1.2); under such kind of premise, the quality is lightened by 63.82%. Moreover, since the structure of the parts becomes a bracket structure after the lightweight design, the shock absorption performance will be greatly improved. The 3D-printed parts have a series of advantages, namely bright surface, low roughness, no obvious warpage and other defects, and good molding effect, which lays solid the foundation for the mass production of high-performance shock-absorbing load-bearing parts.

**Keywords:** 3D printing; shock-absorbing-bearing parts; lightweight; topology optimization; geometric reconstruction

## 1. Introduction

As important means of transportation for people to be engaged in various social activities, motorcycles, electric vehicles, and bicycles, etc. take up a prominent position. Under the pressure of energy, environmental protection, and people's pursuit of high-performing traveling tools, the re-improvement of the properties of traditional mechanical parts has gradually gained attention. The structural strength and material distribution of parts are key factors affecting properties. In the past, subject to the constraint of manufacturing conditions, the development of high-performing mechanical parts was restricted, and the emergence of 3D printing technology created a broad space for the development of new parts. Three-dimensional printing technology is a technology that slices and layers a 3D model, obtains cross-sectional data, and then imports it into rapid prototyping equipment using special software, and manufactures physical parts by superimposing the materials layer by layer. Layer-by-layer superimposition allows 3D printing technology to manufacture parts with nearly any geometry, and allows processing of small batches or single parts and the preparation of finished parts with complex geometries and compact structure [1,2]. Fused deposition modeling (FDM) is a kind of 3D printing technology that heats and melts filamentous hot melt materials and extrudes them out through a sprayer with a tiny nozzle [3,4]. To this end, we made a lightweight design of motorcycle shock-absorbing

and load-bearing components based on 3D printing technology, in the hope of obtaining high-performing motorcycle shock-absorbing and load-bearing components.

Domestically, Wang Lijuan from Chongqing University of Technology [5] optimized a motorcycle frame in Isight software, and reduced the total mass of frame pipes by 7.7% after optimization. Lei Peng et al. [6] analyzed the primary cause of poor vibration comfort of main and auxiliary pedals of a curved beam motorcycle by combining test analysis with computer-aided engineering, and designed a structural optimization scheme to control the vibration of this motorcycle based on relevant analysis results. In 2019, Zhou Zhiqiang [7] from North China University of Water Resources and Electric Power made a strength analysis and modal analysis of the frame of a motorcycle with large discharge capacity, optimized the wall thickness of frame, and finally reduced the weight of frame by 11.35% on the premise of guaranteeing properties. Dai Binzhou [8] carried out lightweight multi-objective optimization on the motorcycle frame using the global response search algorithm, lowered the mass of motorcycle frame by 11.78%, and increased the frequency of the seventh mode by 10.20%, making the possibility of resonance caused by engine excitation very small. In 2020, Wu Nan et al [9]. used finite element analysis software to analyze the trike swing scooter, and further judged the rationality of the structural design of the trike scooter on the basis of strength analysis. Tan Xinxin from South China University of Technology [10] designed and optimized the frame of the electric motorcycle. After the efforts of Yue Liu [11] and his team members, the quality of low-speed electric vehicles has been ultimately reduced by combining a Radial Basis Neural Network (RBF) surrogate model and particle swarm optimization algorithm, based on comprehensive consideration of the performance indicators such as mass, mode, strength, and bending-torsional stiffness. Aiming at being light weight, Yajuan Chen and the team members [12] established optimization variables through sensitivity analysis, and optimized design with strength as a constraint function, and obtained the optimal pipe diameter parameters of motorcycle frame. Using NX Nastran software, Yanjun Xiang [13] and team members built a finite element calculation and analysis model of the electric motorcycle frame, and analyzed three frame stress limit conditions and frame- bending conditions including horizontal working condition, vertical working condition, seat cushion working condition, and the stress and deformation under working conditions. Guojun et al. [14] studied the arrangement of the toes of the mole, reconstructed the toe model with CAD, and finally obtained the toe contour curve, so as to design a bionic sawtooth cutter. Abroad, D Vdovin's team from Moscow State University of Engineering [15] used topology optimization technology to make a lightweight design on the frame of a four-wheeled motorcycle, increased the stiffness of new frame by 2 times, and reduced the weight by 17% in 2019. In 2020, Y Jung et al. from Hyundai Motor of South Korea [16] came up with a multi-material topology optimization technology, achieved different material configurations on different structures, and lowered the body mass of electric buses through the optimized lightweight design. In 2020, Suphanut Kongwat et al. from Shibaura Institute of Technology in Japan [17] designed a lightweight frame with required stiffness based on topology optimization and reduced the mass by 2.5%. Kaveh A et al. [18] and team members used a hybrid algorithm to verify the optimization of the truss structure, showing that the method is more robust and stable than other algorithms. Cho J.G. et al. [19] and his team members selected a CFRP-AL honeycomb sandwich composite material for the lightweight design of the frame, and the weight was reduced by 29.0% after optimization. Symmetry and periodic constraints are introduced to solve the minimum compliance problem by Ballo F et al. team [20], who optimized the wheel by considering several loading conditions. Referring to Li S and team [21], a body optimization design method based on the lightweight target was proposed by analyzing the strength of an all-aluminum body-in-white.

At present, some progress has been made in the lightweight design of a motorcycle frame. The above studies partly improve the overall properties of a motorcycle, but the improvement of the properties of key shock-absorbing and load-bearing components of a

motorcycle needs further research. In this paper, the key shock-absorbing components of a motorcycle will be further optimized based on 3D printing technology.

## 2. Materials and Methods

### 2.1. Design and Analysis Methods

In order to expand the design space of shock-absorbing bearing parts, some pores of shock-absorbing bearing parts need to be filled before lightweight design, as shown in Figure 1a. According to the design requirements, we first use Altair inspire software to analyze the stress of the motorcycle damping components, then carry out topology optimization design and, finally, check the strength. The weight was lowered and the materials were saved as far as possible on the premise of meeting actual stiffness and properties. The most reasonable material distribution was achieved by combining topology optimization with 3D printing, and the optimal properties were realized with the minimum materials. The same simulation parameters were set for force analysis and topology optimization and the specific parameters were as follows: after the parts were imported, the unit was set as mm kg N s and the analysis material were set as ABS, a load of 350 N was applied to position 1 (the direction vectors were −0.70711, 0, and −0.70711), a load of 350 N was applied to position 2 (the direction vectors were −0.70711, 0, and −0.70711), and a load of 350 N was applied to position 3 (the direction vectors were 0.70711, 0, and 0.70711), and a load of 900 N was applied to the negative direction of X, the central position of the connecting line between two holes in Position 4, as shown in Figure 1b. Inertia relief was used for constraint, "more accurate" was selected for calculation speed/accuracy, and single load condition was selected warping the working condition. The size of analysis unit was set to 2 mm, and inertia relief was ticked. Surface treatment was carried out on the motorcycle shock-absorbing and load-bearing components printed by 3D. To begin with, the support was removed, then, rough polishing was performed with sandpaper and, finally, a polishing cloth was adopted for polishing.

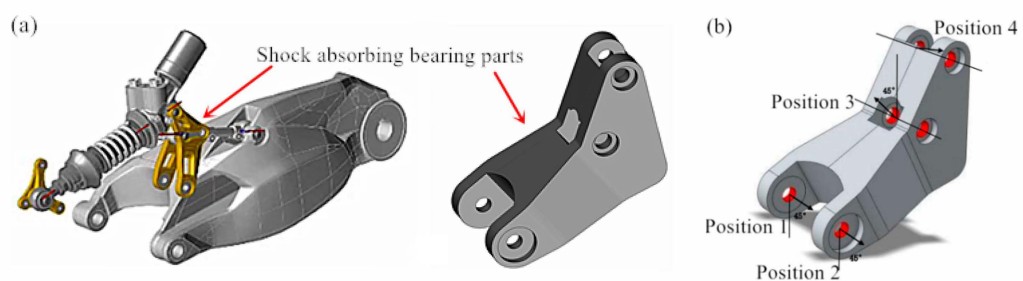

**Figure 1.** Motorcycle shock-absorbing bearing parts: (**a**) Filling effect and (**b**) Restraint and load application position.

### 2.2. Manufacturing Method

Since our design was a redesign of motorcycle shock-absorbing parts, it belonged to a single small-batch parts in the product research and development R&D stage, and the cost can add up considerably if machining, casting, welding, and other methods are adopted. Thus, this paper mainly adopted the 3D printing method to manufacture such kinds of complex parts. To this end, we used an industrial, high-precision desktop 3D printer manufactured by JG Aurora (Shenzhen, China) for trial production. The molding material was ABS (Young's modulus was 2000 MPa, Poisson's ratio was 0.35, the density was 1060 kg/m$^3$, and the yield stress was 45 MPa), the processing layer thickness was set to 0.1 mm, and the filling density was set to 20%.

## 3. Results

### 3.1. Initial Strength of Shock-Absorbing and Load-Bearing Components

Based on the above setting of analysis parameters, the initial model of the shock-absorbing and load-bearing components was imported into Altair Inspire software for initial strength analysis. The results are shown in Figure 2. As shown in Figure 2, the maximum displacement of the motorcycle shock-absorbing and load-bearing components was 0.276 mm and located at Position 4, and the maximum equivalent von Mises stress was 14.67 Mpa and located at Position 3, which did not exceed the yield stress of materials 45 MPa. The minimum safety factor was 3.1 and located at Position 3, which was greater than the minimum 1.2, and the test mass was 258.47 g, which satisfied the requirement of design strength, but there was still much room for improvement in its properties. For this reason, we further optimized the shock-absorbing and load-bearing components in Altair Inspire software (version 2021).

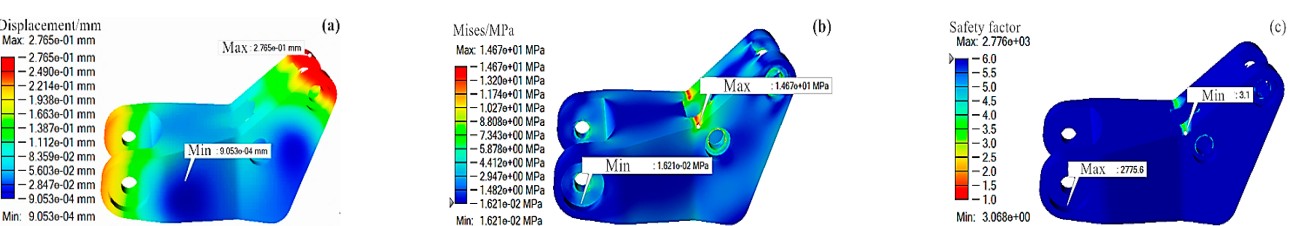

**Figure 2.** Strength analysis of shock-absorbing bearing parts: (**a**) Displacement nephogram; (**b**) Stress nephogram; and (**c**) Safety factor.

### 3.2. Optimization of Shock-Absorbing and Load-Bearing Components

#### 3.2.1. Setting of Topology Optimization

In Altair Inspire software, the parts other than fixed holes of motorcycle shock-absorbing and load-bearing components were designated as the design space, as shown in red in Figure 3, and other parts were non-design space, as shown in gray in Figure 3. To obtain excellent topology optimization results, we set different shape controls for the motorcycle shock-absorbing and load-bearing components, and compared and analyzed the strength analysis results under different shape controls. Since the shape characteristics of this model were not suitable for a one-way draft, the shape control of a one-way draft was not elaborated on here. The setting of other shape constraints is shown in Figure 3. The optimized target mass was set to 30%, the thickness constraint was 3 mm, and inertia relief was ticked.

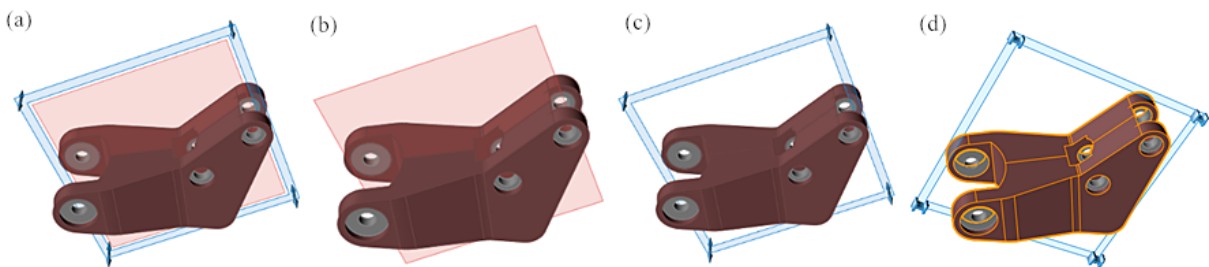

**Figure 3.** Shape control: (**a**) Bidirectional symmetry + draft; (**b**) Symmetry; (**c**) Bidirectional draft; and (**d**) Squeeze out.

#### 3.2.2. Analysis of Topology Optimization Results

The strength analysis results of the shock-absorbing and load-bearing components under different shape controls are shown in Table 1. By observing Table 1 and taking the interaction between the stress distribution, displacement, and safety factor of the shock-absorbing and load-bearing components into account, we found that two-way draft+

symmetry had the best results, which not only satisfied actual needs, but also met the requirements of relevant parameters. After optimization, the mass was low, which was in line with the purpose of lightweight design.

**Table 1.** Strength analysis results of shock-absorbing-bearing parts under different shape controls.

| Shape Control | Optimized Model | Quality/g | Qisplacement/mm | Safety Factor | Equivalent Stress/MPa |
|---|---|---|---|---|---|
| Symmetrical + bidirectional draft | | 80.62 | 0.69 | 1.3 | 35.4 |
| Symmetric | | 87.00 | 0.65 | 2.4 | 67.1 |
| Bidirectional draft | | 89.03 | 0.58 | 1.4 | 32.0 |
| Squeeze | | 82.54 | 0.017 | 1.0 | 45.6 |

### 3.3. Geometric Reconstruction

Through the above analysis, we can learn that two-way draft + symmetry had the best comprehensive properties, which not only satisfied actual needs, but also met the requirements of relevant parameters. After optimization, the mass was low, which was in line with the purpose of lightweight design. Therefore, below we mainly reconstruct the results under the constraints of two-way draft+ symmetry. In Altair Inspire software, there were usually two ways to reconstruct the model: automatic reconstruction and manual reconstruction. Next, we will analyze and compare the results, respectively.

3.3.1. Automatic Reconstruction

After the completion of topology optimization, firstly, the topology optimization results were adjusted to the optimal model by dragging the sliding block, then the option of smoothing result was ticked and POLYNURBS was clicked to fit and obtain the solid model, as shown in Figure 4a. If there was poor contact between the topology optimization model and the fixed hole, we can double-click the model to adjust and then press OK, as shown in Figure 4b. Eventually, a Boolean operation was performed on the reconstructed model to make the design space intersect with the non-design space, and chamfering was performed with a chamfering tool, as shown in Figure 4c. By observing Figure 4c, we can find that the design space and the non-design space of the shock-absorbing and load-bearing components had good contact and smooth transition. The test mass was 80.62 g, 68.8% less than before optimization, and achieved a great weight reduction. In addition, by observing the structure of the shock-absorbing and load-bearing components, we can see that the structure was a supporting structure, which was like an energy storage element. This can greatly improve the shock absorption of shock-absorbing and load-bearing components.

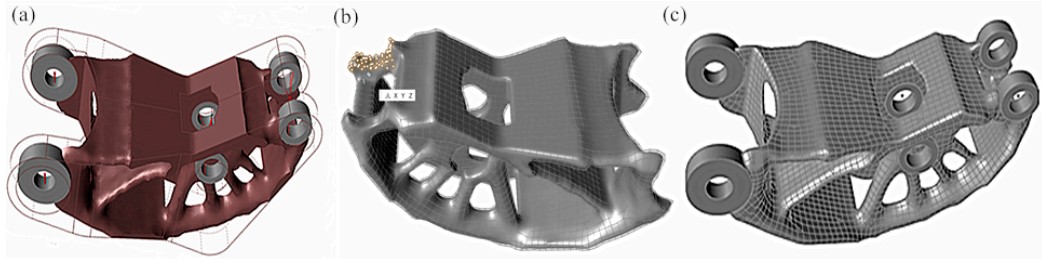

**Figure 4.** Automatic reconstruction: (**a**) Fitting effect; (**b**) Local adjustment; and (**c**) Reconstruction effect.

### 3.3.2. Manual Reconstruction

Manual reconstruction adjusted the optimal topology by adjusting the sliding bar and density factor of optimization results, then carried out manual fitting for geometric reconstruction, and obtained better results by adjusting the position of control point. Manual reconstruction began by creating a POLYNURBS block at the position of installation hole, and then wrapping the part through the wrap command, as shown in Figure 5a. A hole was drilled at the fixed hole of the wrapped part, as shown in Figure 5b. Eventually, OK was clicked to get the final result of manual reconstruction, as shown in Figure 5c. By observing Figure 5c, we can see that the manually reconstructed model was smoother than the automatically reconstructed model, and full wrapping was realized at the fixed hole position, which can make the force on the parts more uniform. In the process of reconstruction, it was found that there was sheet body in the reconstructed model, so a Boolean operation cannot proceed. After the analysis and discussion by the team, it was considered that the non-design space was an imported model and must be an entity, and a sheet body only occurred in the reconstructed part, indicating that there were errors in the reconstructed results during parameter setting or reconstruction, which resulted in the generation of a sheet body and errors. Conclusively, we eliminated the intersection by fine-tuning intersecting facets, improved the optimization results by dragging and adjusting the density factor, etc., and finally got better geometrically reconstructed parts. Next, we will analyze the strength of two reconstructed models and compare their properties.

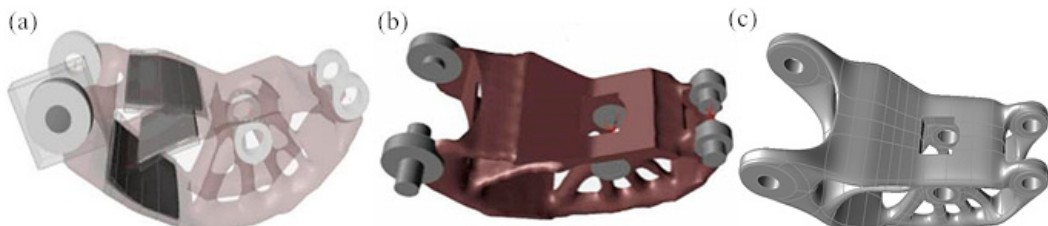

**Figure 5.** Manual reconstruction: (**a**) Model package; (**b**) Opening of fixed holes; and (**c**) Reconstruction effect.

### 3.4. Strength Check of the Parts with Optimized Topology

Shock-absorbing and load-bearing components reconstructed automatically and manually were imported into Inspire software, respectively, and the strength was checked again after the same simulation parameters as those in the part strength analysis before topology optimization were set. The results are shown in Table 2. By comparing the manual and automatic reconstruction results in Table 2, we can see that compared with automatic reconstruction, manual reconstruction had a smoother surface, better pliability, smaller displacement (the maximum displacement decreased by 20.8%), smaller stress concentration (the maximum stress decreased by 15.4%), a higher safety factor (increased by 13.3%), and slightly increased weight (increased by 15.9%), but on the whole, manual reconstruction worked better.

**Table 2.** Comparison of automatic and manual reconstruction optimization results.

| Reconstruction Mode | Optimized Model | Quality/g | Displacement/mm | Safety Factor | Equivalent Stress/Mpa |
|---|---|---|---|---|---|
| Automatic | | 80.62 | | | |
| | | | 0.69 | 1.3 | 35.4 |
| Manual | | 93.49 | | | |
| | | | 0.55 | 1.5 | 29.9 |

The strength check results of the motorcycle shock-absorbing and load-bearing components after lightweight design are shown in Figure 6. By observing Figure 6, we can see that after lightweight design, the maximum equivalent von Mises stress of the shock-absorbing and load-bearing components was 29.99 Mpa, the maximum displacement was 0.54 mm, the minimum safety factor was 1.5, and the mass was 93.49 g. By comparing the initial strength analysis results of the shock-absorbing and load-bearing components, we can find that the maximum equivalent von Mises stress was 14.59 Mpa, the maximum displacement was 0.27 mm, the minimum safety factor was 3.1, and the mass was 258.47 g. The lightweight design of the motorcycle structural components reduced the weight by 63.82%. At the same time, the maximum equivalent von Mises stress (29.99 Mpa) of components did not exceed the yield stress (45 MPa) of materials, and the safety factor (1.5) was greater than the minimum safety factor (1.2), which satisfied the actual strength requirements.

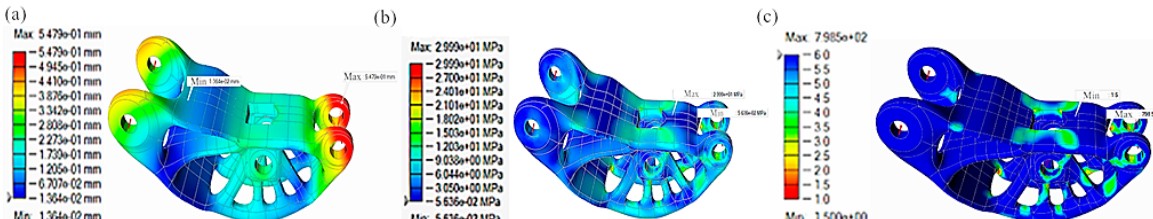

**Figure 6.** Strength analysis of shock-absorbing-bearing parts: (**a**) Displacement nephogram; (**b**) Stress nephogram; and (**c**) Safety factor.

*3.5. D Printing of Shock-Absorbing and Load-Bearing Components*

3.5.1. Setting of Print Parameters

First of all, the original model and the manually reconstructed model were saved in stl format, and then imported into UP Studio 3, a 3D printing data processing software, to set printing parameters. The specific parameters were: the nozzle diameter was 0.4 mm, the material was ABS, the printing direction was set, and the support addition was optimized by adjusting the placement. Dynamic layer thickness was adopted. Since the lower part of the model had a large overhang, the layer thickness of the lower half of model was set to 0.15 mm. The upper part of the model was found to have no large overhang. In order to increase the molding speed, the layer thickness of the upper part of the model can be raised to 0.2 mm, the filling density was set to 25%, the support density to 20%, the filling path to ZigZag, the filling angle to 45°, the angle growth to 90, the filling growth to 0.00, the contour/filling overlap to 0.20 mm, the bottom overlap to 0.20 mm, and the printing speed to fine. After parameter setting, the model is shown in Figure 7.

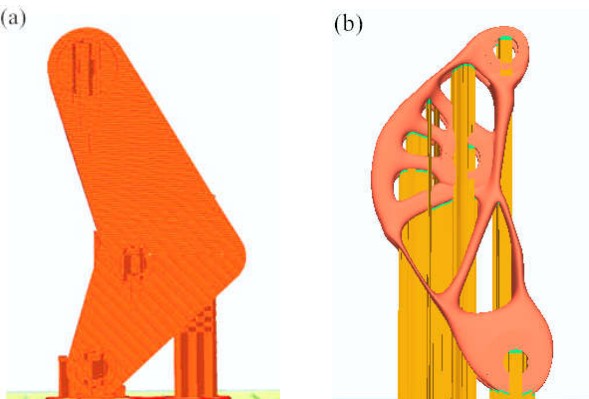

**Figure 7.** Effect of data processing and support addition: (**a**) Original model and (**b**) Manual reconstruction model.

### 3.5.2. Analysis of 3D Printing Effect

The final model effect of 3D printing is shown in Figure 8. By observing Figure 8, we can see that the printed model had a bright surface and low roughness, without obvious slag on the overhang, obvious warpage, and other defects on the shock-absorbing and load-bearing components. Certain support additions at fixed holes and other positions may affect surface smoothness, but within the allowable range, as shown in Figure 8a,b. The printed parts were removed from the substrate, and the final part model was obtained after removing the support and polishing, grinding, and removing surface burrs, etc., as shown in Figure 8c,d.

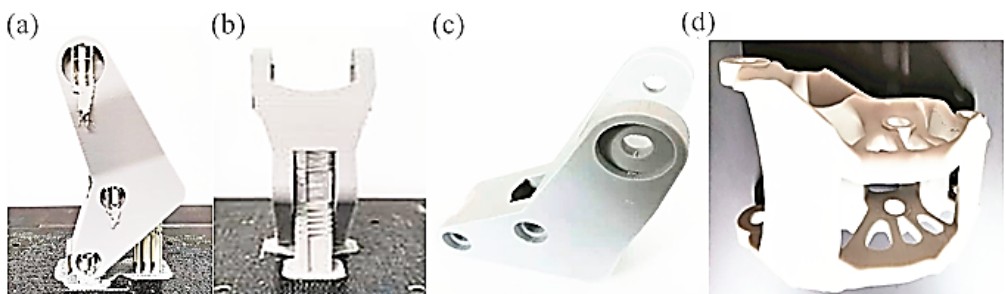

**Figure 8.** Printing effect: (**a**) Original model; (**b**) Topology optimization model: (**c**) Original model; and (**d**) Topology optimization model.

### 4. Discussion

Based on the above results, we can see the lightweight shock-absorbing and load-bearing components designs based on 3D printing technology basically fulfill the expected goal, greatly reduce weight, and realize the uniform distribution of stress on the premise of ensuring part safety. However, some details should be noticed during model design and 3D printing: (1) in the process of model reconstruction, the safety factor of model often fails to meet requirements. After discussion and analysis by the team, it was found that one of the reasons is that some positions of the parts are not chamfered or the degree of chamfering is not enough, so in the process of model reconstruction, we must set aside sufficient chamfering at the contact position between the design space and non-design space. Another major reason is that after geometric reconstruction, the part is not tightly coupled with the non-design space, making the overall safety factor too low. The solution is to stretch the contact between the geometric reconstruction part and the non-design space part, as shown in Figure 9a. (2) There are intersecting facets inside shock-absorbing and load-bearing components during manual fitting. Through analysis, this is induced by the overadjustment of surface control points of shock-absorbing and load-bearing components

during reconstruction. Later, the control points are fine-tuned during redesign and the adjacent facets are subjected to linkage adjustment, so that there is no sheet body in the reconstruction body, and a good design effect can be achieved, as shown in Figure 9b. (3) In the shock-absorbing and load-bearing components printed by 3D, there is substrate warpage. By analyzing the cause of warpage, we can find that warpage is mainly affected by the temperature of the bottom plate, the height of the nozzle, and the levelness of the printing platform, etc. The solution is to increase the molding quality by setting the temperature of the bottom plate to 50 °C, adjusting the height of nozzle, raising the adhesion between the substrate and platform, and calibrating the levelness of the printing platform, etc. Moreover, if the layer thickness of 3D printing is not properly set, ripples will emerge. It was also found that the surface quality can be improved by reducing the layer thickness appropriately, as shown in Figure 9c.

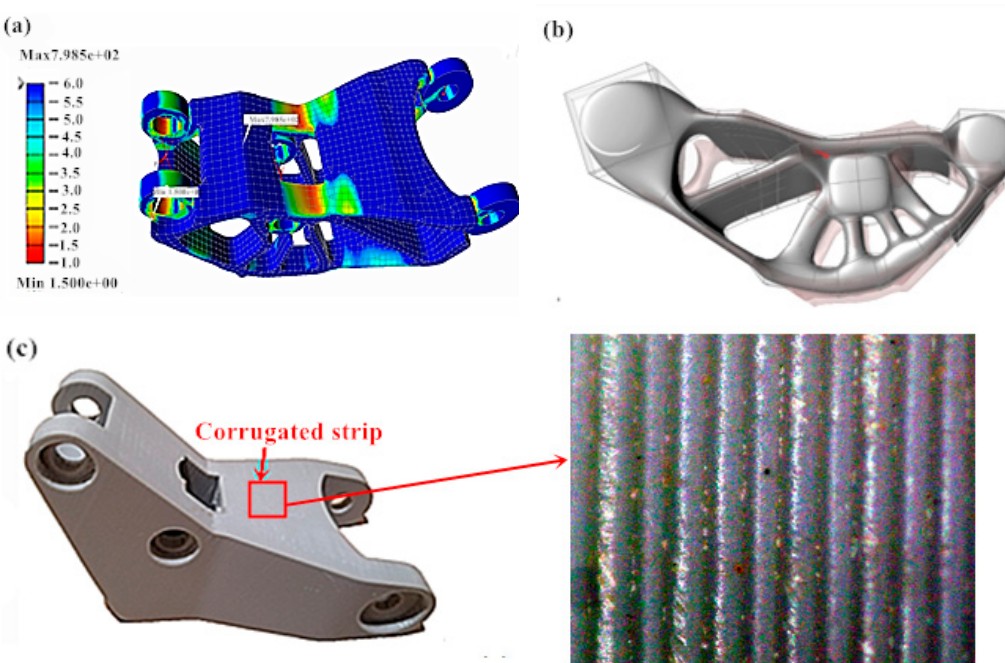

**Figure 9.** Analysis of design and manufacturing problems: (**a**) Too small safety factor; (**b**) Intersection phenomenon; and (**c**) Unreasonable layer thickness.

## 5. Conclusions

(1)  The initial strength analysis results of the shock-absorbing and load-bearing components are as follows: the maximum equivalent von Mises stress is 14.59 Mpa, the maximum displacement is 0.27 mm, the minimum safety factor is 3.1, and the mass is 258.47 g.

(2)  After lightweight design, the strength check results of the shock-absorbing and load-bearing components are: the maximum equivalent von Mises stress is 29.99 MPa, the maximum displacement is 0.54 mm, and the minimum safety factor is 1.5. The lightweight design of the shock-absorbing and load-bearing components reduces the weight by 63.82%, and ensures that the maximum equivalent von Mises stress of part does not exceed the yield stress of materials (45 MPa), and the safety factor (1.5) is greater than the minimum safety factor (1.2), which meets the actual strength requirements.

(3)  The shock-absorbing and load-bearing components printed by 3D technology has a bright surface and low roughness, without obvious warpage and other defects, and the molding effect is good.

Definitely, to further improve the comprehensive properties of shock-absorbing and load-bearing components printed by 3D technology, we still need subsequent experiments—for example, the design of full-grid shock-absorbing and load-bearing components with higher shock absorption. This study lays a foundation for the design and batch manufacture of high-performing shock-absorbing and load-bearing components.

**Author Contributions:** G.Z. and J.L. completed the design of parts. X.Z. and Y.Z. completed the manufacturing. A.W. and R.F. completed the analysis of parts. All authors have read and agreed to the published version of the manuscript.

**Funding:** The study was funded by the Henan Provincial Science and Technology Project (Grant No. 212102310859) and the Key Scientific Research Projects of Colleges and Universities in Henan Province (Grant No. 22A460006). In addition, this work was supported by the Analytical and Testing Center of ZKNUC for carrying out microscopic analysis.

**Institutional Review Board Statement:** Not applicable.

**Informed Consent Statement:** Not applicable.

**Data Availability Statement:** Data sharing is not applicable to this article.

**Conflicts of Interest:** The authors declare no conflict of interest.

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
