# Peer review of "Lightweight Design of Shock-Absorbing and Load-Bearing Components Based on 3D Printing Technology"

_coatings, doi:10.3390/coatings12060833_

Round 1

Reviewer 1 Report

The paper is focused on the analysis of the shock-absorbing and load-bearing  components using Altar Inspire software. The topic could be interesting for readers of Coatings. On this basis, I recommend the publication after the following minor revisions:

  • Tables 1,2. Is it possible to estimate the errors for all the parameters?
  • Abstract and Conclusions should be revised to better highlight the novelty of this paper compared to literature.

Author Response

Point 1: The paper is focused on the analysis of the shock-absorbing and load-bearing  components using Altar Inspire software. The topic could be interesting for readers of Coatings. On this basis, I recommend the publication after the following minor revisions:

Tables 1,2. Is it possible to estimate the errors for all the parameters?

Abstract and Conclusions should be revised to better highlight the novelty of this paper compared to literature.

Response 1: Due to the complexity of the model, the finite element simulation results in Table 1 and table 2 are difficult to verify in practice. There may be some errors between the simulation results and the actual results, but this does not affect the comparison of results under the same conditions. In order to highlight the novelty of this, we have supplemented references 11 / 12 / 13 / 17 / 18 / 19 / 20. We revised the introduction and abstract to highlight innovation.

Reviewer 2 Report

  1. Literature part needs revision like Wang Lijuan FROM Chongqing University of Technology, Zhou 51 Zhiqiang [7] from North China University, Tan Xinxin from South China University of 61 Technology: What is the purpose of adding institute here?
  2. Why this paper is necessary, how and on what basis the gap in literature/novelty of the present work is not provided properly
  3. "we needed to fill some lightweight hole". "finally checked the strength" Language needs substantial revision
  4. Manufacturing Method: Authors should specify how the parts are manufactured clearly
  5. Meshing, Loading conditions and Boundary conditions are not provided
  6. Convergence, element details and how authors confirmed the results are not provided in this paper, which makes the no sense
  7. Therefore, below we mainly reconstruct the results under the constraints of two-way draft+ symmetry. Check the language
  8. Next, we will analyze and compare the results respectively. Needs revision
  9. Results are not validated before going for 3D printing. How authors confirm the software results and methodology adopted are correct?
  10. Overall, paper needs extensive revision in language, methodology, conclusion part.
  11. Only 13 papers referred, Kindly try to provide more related articles.

Author Response

Point 1: Literature part needs revision like Wang Lijuan FROM Chongqing University of Technology, Zhou 51 Zhiqiang [7] from North China University, Tan Xinxin from South China University of 61 Technology: What is the purpose of adding institute here?

Why this paper is necessary, how and on what basis the gap in literature/novelty of the present work is not provided properly

Response 1: At the end of the introduction, we describe the shortcomings of the current research and the necessity of this research

Point 2: "we needed to fill some lightweight hole". "finally checked the strength" Language needs substantial revision

Response 2: We revised the above language problems and checked the full text

Point 3: Manufacturing Method: Authors should specify how the parts are manufactured clearly

Response 3: We have described the manufacturing method in detail in part 2.2 of the paper.

Point 4: Meshing, Loading conditions and Boundary conditions are not provided

Convergence, element details and how authors confirmed the results are not provided in this paper, which makes the no sense

Response 4: We provide loads and boundary conditions in Figure 1 and part 2.1 of the paper. In part 2.1 of the paper, we provide the unit size and so on. If the analysis does not converge, the analysis will fail.

Point 5: Next, we will analyze and compare the results respectively. Needs revision

Results are not validated before going for 3D printing. How authors confirm the software results and methodology adopted are correct? Overall, paper needs extensive revision in language, methodology, conclusion part.

Response5: We revised the above language problems and checked the full text. Software analysis can only provide basis for actual manufacturing and cannot be determined before 3D printing.

Point 6: Only 13 papers referred, Kindly try to provide more related articles.

Response6 : We supplemented references 11 / 12 / 13 / 17 / 18 / 19 / 20.

Reviewer 3 Report

The Manuscript is aimed to redesign some shock-absorbing and load-bearing components using modeling software. Topological optimization was provided taking into account 3D-printing capabilities. The manuscript contains some interesting results and could be interesting for practice-oriented scientists and engineers. However, there is significant revision of all sections is needed to improve scientific soundness, to make manuscript more clear and logical and to close the gaps in provided information. There are some main comments are listed below.

  • Lines 9-11: repeating of words in one sentence (…shock-absorbing and load-bearing…(…excellent shock-absorbing…))…to redesign shock absorbing and load-bearing components…
  • Lines 13-14: I recommend to add information about what 3D-printing method was employed along with what material was used within the study. Now it is unclear, revision of the Abstract is needed.
  • Introduction section should be also totally revised. Now it has only 13 references and discussed information does not give clear image of the state of the art. As far as the Manuscript represents not short letter but full-length article, the list of references should be expanded at list to 25-30 items. Moreover, no deep and comparative analysis of the background is presented, presentation of information and style make Introduction section unclear for readers.
  • As far as authors deal with ABS material at least brief information about that material and its properties should be given in Introduction section.
  • Section 2.2: There is still no information about what 3D-printing methods was used for experiments. The related details should be provided including synthesis parameters and more information about feedstock material.
  • Section 3: Only the name of used software (Altair Inspire) is mentioned in the Manuscript. There is no details about mathematical model applied for modeling (which parameters of part and material are taken into account and which are not). How was topological optimization performed, is there only weight was used as minimization parameter or not? More detailed discussion is needed.]
  • The description about which buttons were clicked in software does not conform the scientific soundness and style implied when publishing in a scientific journal. The manuscript should be revised also from that point of view.
  • Lines 231-235: Provided information about 3D-printing should be moved to Materials and methods section.

Based on above, major and deep revision of the Manuscript is needed paying a lot of attention to improvement of scientific soundness and style.

Author Response

Point 1:  Lines 9-11: repeating of words in one sentence (…shock-absorbing and load-bearing

…(…excellent shock-absorbing…))…to redesign shock absorbing and load-bearing components…

Response 1: We have carefully revised the abstract and introduction.

Point 2: Lines 13-14: I recommend to add information about what 3D-printing method was employed along with what material was used within the study. Now it is unclear, revision of the Abstract is needed.

Response 2: We added the method of 3D printing parts in part 2.2 of the paper.

Point 3: Introduction section should be also totally revised. Now it has only 13 references and discussed information does not give clear image of the state of the art. As far as the Manuscript represents not short letter but full-length article, the list of references should be expanded at list to 25-30 items. Moreover, no deep and comparative analysis of the background is presented, presentation of information and style make Introduction section unclear for readers.

Response 3: We supplemented references 11 / 12 / 13 / 17 / 18 / 19 / 20.

Point 4: As far as authors deal with ABS material at least brief information about that material and its properties should be given in Introduction section.

Response 4:  We give the details of ABS materials in part 2.2 of the paper.

Point 5: Section 2.2: There is still no information about what 3D-printing methods was used for experiments. The related details should be provided including synthesis parameters and more information about feedstock material.

Response 5: In part 2.2 of the paper, we give the detailed information of 3D printing parameters and ABS materials.

Point 6: Section 3: Only the name of used software (Altair Inspire) is mentioned in the Manuscript. There is no details about mathematical model applied for modeling (which parameters of part and material are taken into account and which are not). How was topological optimization performed, is there only weight was used as minimization parameter or not? More detailed discussion is needed.]

Response 6: Because the initial structure of the part is relatively simple, it can be completed by using general basic software, and the technical requirements are not high, so there is not much introduction.The specific parameters of topology optimization are described in detail in sections 2.1 and 2.2.1.

Point 7: The description about which buttons were clicked in software does not conform the scientific soundness and style implied when publishing in a scientific journal. The manuscript should be revised also from that point of view.

Response 7: We have revised the corresponding part.

Point 8: Lines 231-235: Provided information about 3D-printing should be moved to Materials and methods section.

Response 8: The material and method part introduces the specific parameters and materials of 3D printing. Part 3.5.1 of the paper is the optimization result of part support addition, and there is no conflict between the two.

Round 2

Reviewer 2 Report

Nil

Author Response

Thank you very much for your approval.

Reviewer 3 Report

Authors made considerable revision of the manuscript taking into account given comments and suggestions. The manuscript is becoming more logical and consistent.

However I'm still not sutisfied with a number of things which sholud be cleared and revised to improve scientific soundness and valuability of the study.

1. The Introduction section still does not correspond a style requirements of scientific journal. The background analysis and the reference list still look insufficient.  I reccomend to revise it once again.

2. Since initial part structure is simple and using general basic software is enough what how could you explain and justify value and significance of obtained results? If the provided experiment is just simple modeling by means of standard software instruments scientific sence of the study is not clear at all.

Author Response

Point 1: The Introduction section still does not correspond a style requirements of scientific journal. The background analysis and the reference list still look insufficient.  I reccomend to revise it once again.

Response 1: We have supplementary references 14

Point 2: Since initial part structure is simple and using general basic software is enough what how could you explain and justify value and significance of obtained results? If the provided experiment is just simple modeling by means of standard software instruments scientific sence of the study is not clear at all.

Response 2: The initial design of parts adopts general three-dimensional software, but the optimization design of parts adopts special finite element analysis software. The focus of this paper is the topology optimization design based on 3D printing technology.